# Global ground strike point characteristics in negative downward lightning flashes — part 2: Algorithm validation

Dieter R. Poelman[1], Wolfgang Schulz[2], Stephane Pedeboy[3], Leandro Z. S. Campos[4,5], Michihiro Matsui[6], Dustin Hill[7], Marcelo Saba[8], Hugh Hunt[9]

[1]Royal Meteorological Institute of Belgium, Brussels, Belgium

[2]Austrian Lightning Detection and Information System (ALDIS), Vienna, Austria

[3]Météorage, Pau, France

[4]Campos Scientific Computing, São José dos Campos, Brazil

[5]Currently a consultant for Andela Inc., New York, NY, USA

[6]Franklin Japan Corporation, Sagamihara 212-0212, Japan

[7]Scientific Lightning Solutions LLC (SLS), Titusville, Florida, USA

[8]National Institute for Space Research, INPE, São José dos Campos, Brazil

[9]The Johannesburg Lightning Research Laboratory, School of Electrical and Information Engineering, University of Witwatersrand Johannesburg, Johannesburg, South Africa

*Correspondence to*: Dieter R. Poelman (dieter.poelman@meteo.be)

**Abstract.** At present the lightning flash density is a key input parameter to assess the risk of occurrence of a lightning strike in a particular region of interest. Since it is known that flashes tend to have more than one ground termination point on average, the use of ground strike point densities as opposed to flash densities is more appropriate. Lightning location systems (LLSs) do not directly provide ground strike point densities. However, ingesting their observations into an algorithm that groups strokes in respective ground strike points results in the sought after density value. The aim of this study is to assess the ability of three distinct ground strike point algorithms to correctly determine the observed ground-truth strike points. The output of the algorithms is tested against a large set of ground-truth observations taken from different regions around the world, including Austria, Brazil, France, Spain, South Africa and the United States of America. These observations are linked to the observations made by local a LLS in order to retrieve the necessary parameters of each lightning discharge and serves as input for the algorithms. Median values of the separation distance between the first stroke in the flash and subsequent ground strike points is found to vary between 1.3 km and 2.75 km. It follows that all three of the algorithms perform well, with success rates up to about 90% to retrieve the correct type of the strokes in the flash, i.e., whether the stroke creates a new termination point or follows a pre-existing channel. The most important factor that influences the algorithms' performance is the accuracy by which the strokes are located by the LLS. Additionally, it is shown that the strokes' peak current plays an important role, whereby strokes with a larger absolute peak current have a higher probability of being correctly classified compared to the weaker strokes.

# 1 Introduction

Severe weather has always been around. However, its global impact on both society and economies increases steadily, with no signs of decline whatsoever towards the future. More specifically, the deleterious effects of lightning discharges should not be underestimated. In this respect, cloud-to-ground (CG) flashes play a particular part since they have an enormous impact on nature and society, both directly and indirectly. Besides lightning-caused fatalities and injuries that are reported each year worldwide (Curran et al. 2000; Holle et al. 2005, 2016), it is a well-known fact that lightning is a major cause of, for example, wildfires when the conditions to ignite fire near the vicinity of the ground strike point are fulfilled (Balch et al., 2017; Williams, 2016; Schultz et al., 2019). On the other hand, the economic effects of lightning damage to property are immense, whether being an individual household or a large-sized company, with total costs that quickly can spiral out of control. In this matter, electrical appliances are vulnerable to the electromagnetic fields induced by lightning. Additionally, the search for alternative ways of generating energy has led to the construction of vast amounts of wind turbines and wind and solar farms all over the world, to name but one other example. However, it has been demonstrated by Montanyà et al. (2014) by analyzing Local Mapping Array (LMA) observations in Spain that the rotating blades of wind turbines can trigger lightning, thereby causing self-induced damages. Not to mention the detrimental effects of lightning in other areas such as aviation, it is clear that adequate lightning protection measures need to be put at place to mitigate the effects of lightning impacts. For a comprehensive overview of lightning hazards to human societies the interested reader is referred to Koshak et al. (2015) and Yair (2018).

Over the years, our knowledge of thunderstorms has greatly improved, not least in the field of lightning. By means of high-speed cameras, it has been observed that roughly half of the downward negative CG multiple stroke flashes exhibit more than one ground strike point (GSP). With an average value varying around 1.5 to 1.7 GSPs per flash (Rakov et al., 1994; Hermant, 2000; Valine and Krider, 2002; Saraiva et al., 2010, Poelman et al., nhess-12-2021 companion paper). This implies that the average number of lightning strike points is about 50% to 70% higher than the observed number of flashes. Additionally, the distance between the different GSPs and the first stroke in the flash is of the order of a few kilometers (Thottappillil et al., 1992; Valine et al., 2002; Stall et al., 2009). It follows that every ground strike point is a potential threat and therefore ground strike points ought to be taken into account when it comes to lightning risk estimation for lightning protection.

Nowadays the primary input parameter in lightning risk assessment applications is the lightning flash density, $N_G$. The latter is defined as the number of CG flashes per square kilometer per year. In the past, an empirical formula was applied to infer $N_G$ from the keraunic level of thunderstorm days. However, progress made over the years to detect lightning discharges by means of lightning location systems (LLSs) has led to $N_G$ being determined from the ground flash measurements by LLSs. By definition, the location of a flash has historically been determined by that of the first stroke in the flash; while some LLSs use the centroid of the strokes' locations. Taking into account that on average more than one GSP is observed per flash, it follows that the use of $N_G$ in the risk calculation of lightning protection leads to an underestimation of the hazard. It is for this reason that $N_G$ should be replaced by the lightning strike point density. Nowadays LLSs provide stroke locations with median accuracies in the order of a few hundred meters or better, hence LLSs can provide strike point densities after applying a

dedicated algorithm to group the individual strokes within a flash in ground strike points. This is in particular helpful to further improve the risk estimation for lightning protection since it is derived from the density of lightning ground strike points in a region.

In this study, three different ground strike point algorithms are tested against a large set of high-speed video measurement data from multiple regions to find out their ability to determine the observed ground strike points correctly. In Sections 2 and 3 the
different lightning location systems and ground-truth data sets are described, respectively, followed by the characteristics of the algorithms in Section 4. In Section 5 the results are discussed, while Section 6 summarizes the study and draws some further conclusions.

## 2. Lightning Location Systems involved

The ground-truth data sets outlined in Poelman et al. (nhess-12-2021, companion paper) and gathered in Austria (AT) in 2012,
2015, 2017 and 2018, Brazil (BR) in 2008, South-Africa (SA) in 2017-2019, and the United States of America (US) in 2015, serve among others as input for the ground strike point algorithms described further in Sec. 3. In addition, two extra ground-truth data sets collected in France (FR) during 2013-2016 and Spain (ES) in 2017-2018 are included in this study. Whereas the flash grouping is based on the high-speed video images, the information of, e.g., location, peak current and semi-major axis of the 50% confidence ellipse, is retrieved by linking the ground truth data to the observations made by a local ground-
based LLS. In this Section, the different LLSs are briefly described.

### 2.1. ALDIS

ALDIS operates a sensor network of eight low frequency (LF) lightning detection sensors in Austria while the central processor ingests additional sensors from neighbouring countries. In addition, ALDIS is partly known for its continuous work related to the European Cooperation for Lightning Detection (EUCLID); recognized as one of the best documented networks in Europe
in terms of location accuracy (LA) and detection efficiency (DE) estimates. This is made possible partly due to the observations made at the instrumented Gaisberg Tower in Austria and supplemented by mobile video and field recording system (VFRS) observations in Austria, as well as throughout Europe. Due to continuous adaptation and improvement of the system with on-going hard- and software upgrades, the median LA is in the range of 100 m (for more detailed information see Schulz et al., 2016; Poelman et al., 2016; Diendorfer, 2016).

### 2.2. Météorage

The French national LLS has been operated by Météorage (MTRG) since 1986. It detects low-frequency electromagnetic signals generated by CG lightning, as well as a fraction of large amplitude intracloud discharges much in the same way as ALDIS. In the beginning, the LLS was made up of sensors placed only in France. Over the years this core network expanded with compatible sensors of neighbouring partners, providing seamless extended observation coverage over western Europe. In

this study, the LLS of MTRG is used to match the ground-truth observations taken in France and Spain. Similar DE and LA values as the ones stated above for OVE-ALDIS are applicable for this network.

### 2.3. RINDAT

At the time, the ground-truth observations used in this work were carried out, the Brazilian Lightning Detection Network (RINDAT) was composed out of a mix of 47 sensors. The network has evolved somewhat since then resulting in an improved
network performance. Nevertheless, a stroke and flash DE of RINDAT of respectively of 55% and 87% was reported by Ballarotti et al. (2006). Additionally, an upper limit on the LA was retrieved of about 5 km. More information on the characteristics of the network is given by Naccarato and Pinto [2009].

### 2.4. SALDN

The South African Lightning Detection Network (SALDN) was first installed in South Africa in 2006 by the South African
Weather Services (SAWS), originally consisting of 19 Vaisala LS7000 sensors spread across the country. The network has since been upgraded to 24 sensors across the country with an average sensor baseline of approximately 150 km, forming a grid across the country (Gijben, 2012; Evert, 2017). Self-evaluation of the network estimates flash detection efficiencies above 90 % and location accuracies within 500 m for all of the coverage of the country, only dropping below these levels at the borders (of the country and the network). Ground-truth evaluations report cloud-to-ground stroke detection efficiencies of 85-
90 %. These evaluations further indicate a median location error within 150 m (Hunt, 2014, Fensham, 2018, Hunt, 2020).

### 2.5. NLDN

The U.S. national lightning detection network (NLDN) adopts a combination of time-of-arrival and direction finding technology (Cummins and Murphy 2009), similar to the other networks, to geolocate lightning CG strokes and IC pulses since 1989. The Contiguous United States (CONUS) is covered by approximately 100 LS7002 sensors (Nag et al., 2014). The
detection efficiency and location accuracy of the NLDN has been evaluated thoroughly using video observations (Biagi et al. 2007; Warner et al 2012; Cummins et al. 2014; Zhang et al. 2015; Zhu et al. 2016), tower data (Lafkovici et al. 2006; Cramer and Cummins 2014; Zhu et al. 2020) and triggered lightning data (Jerauld et al. 2005; Nag et al. 2011; Mallick et al. 2014). It follows that the flash DE is expected to be in the order of 95% within CONUS. The location accuracy is approximately 150 to 250 m over the majority of the United States, and decreasing somewhat to 250-500 m toward the edges of the network.

**3. Data sets**

Since not all of the strokes observed by the high-speed cameras and electric field change sensors were detected by the different LLSs, the data sets used in this study differ slightly from the ones presented in Poelman et al. (nhess-2021-12,

**Table 1. Data set characteristics for Austria (AT), Brazil (BR), France (FR), South Africa (SA), Spain (ES) and the United States of America (US)**

| Parameter | LLS | | | | | |
|---|---|---|---|---|---|---|
| | AT | BR | FR | SA | ES | US |
| $N$(flashes) | 474 | 110 | 354 | 392 | 76 | 73 |
| $N$(strokes) | 1373 | 383 | 894 | 1174 | 183 | 273 |
| $N$(GSP) | 808 | 189 | 585 | 508 | 121 | 114 |
| Location Accuracy | | | | | | |
| Sample Size | 582 | 210 | 325 | 689 | 63 | 161 |
| Mean (km) | 0.38 | 1.88 | 0.73 | 0.65 | 0.37 | 0.67 |
| Median (km) | 0.11 | 1.0 | 0.19 | 0.11 | 0.11 | 0.13 |
| 95th percentile (km) | 1.76 | 6.74 | 3.82 | 2.06 | 1.43 | 4.15 |
| Semi-major Axis | | | | | | |
| Mean (km) | 0.31 | 0.69 | 0.30 | 0.36 | 0.17 | 0.43 |
| Median (km) | 0.08 | 0.50 | 0.20 | 0.20 | 0.15 | 0.20 |
| 95th percentile (km) | 1.43 | 1.66 | 0.80 | 1.50 | 0.33 | 1.10 |
| Resolution provided by LLS (m) | 2012: 100 2015-2018: 10 | 100 | 2013-2015: 100 2016: 10 | 100 | 10 | 100 |
| $\chi^2$ | | | | | | |
| Mean | 1.01 | 4.11 | 1.35 | 0.67 | 1.07 | 1.23 |
| Percentage > 5 | 0.87 | 21.88 | 1.01 | 0.51 | 0 | 2.21 |
| Median absolute peak current (kA) | | | | | | |
| 1st strokes | 12.4 | 19.7 | 15.6 | 18.0 | 11.9 | 31.4 |
| Subsequent strokes | 10.1 | 15.4 | 13.3 | 13.0 | 11.2 | 16.4 |
| NGC | 12.4 | 18.8 | 14.7 | 18.0 | 11.5 | 27.5 |
| PEC | 8.3 | 14.8 | 12.8 | 12.0 | 11.3 | 14.3 |
| Distance between GSP and 1st stroke in the flash | | | | | | |
| Sample size | 334 | 79 | 231 | 116 | 45 | 41 |
| Mean (km) | 2.42 | 3.03 | 2.43 | 3.73 | 2.84 | 1.48 |
| Median (km) | 2.05 | 2.75 | 2.19 | 2.27 | 2.51 | 1.30 |
| 99th percentile (km) | 9.52 | 7.62 | 7.21 | 20.59 | 6.34 | 4.8 |
| Maximum (km) | 16.5 | 8.09 | 13.69 | 20.9 | 6.75 | 5.43 |

companion paper). Note that the list of flashes to test the performance of the ground strike point algorithms is additionally enlarged by two extra data sets gathered in France and Spain. The quality of the latter two data sets is of the same level as compared to the data sets introduced in Poelman et al. (2021, nhess-2021-12, companion paper). However, the limited video recording time of 500 ms prohibits its use in Poelman et al. (2021, nhess-2021-12, companion paper). It should be pointed out that a flash is completely removed if a stroke that creates a new GSP is not detected by the LLS since this would impact the success rate of the algorithm further described in Section 5. In what follows, some of the characteristics of the reduced data sets are discussed. Notice that detection efficiency projections of the LLS are out of the scope of this study and therefore detailed investigation is disregarded as such. Nevertheless, one can find in Section 2 references for the individual LLS detection efficiency estimations of the individual networks.

Some of the characteristics that play a role in the further course of the study are listed in Table 1 for the different data sets and are described in the text that follows. The combined data sets include a total of 1479 flashes, consisting of 4280 strokes, whereby a total of 2325 ground strike points are distributed among them. The size of the data sets, in terms of flashes, strokes and ground strike points, are somewhat smaller compared to Poelman et al. (2021, companion paper) for the reason described above. Because of this, it is not possible and not valid to use the numbers given in Table 1 for detection efficiency estimations. The random location errors of the different LLSs can be quantified by using the strokes that follow the same channel as observed from the consecutive high-speed images. Since those strokes are assumed to strike ground at the same point, the differences between the stroke positions within a GSP lead to the LA estimation after applying a downscaling factor of $\sqrt{2}$. The latter scaling is applied since both positions are subject to random errors, by analogy of Schulz et al. (2010) and Biagi et al. (2007). The differences determined by this method should be regarded as upper bounds of the actual position differences because there is the possibility that the channel geometry and/or the actual ground contact varied slightly from stroke to stroke and was not resolved by the camera. The results hereof can be consulted in Table 1. All of the LLSs have median LAs in the range of 0.11-0.19 km, except for Brazil with a median LA of 1 km. These LA values correspond with previous LA estimates in other studies mentioned in Section 2 for the individual networks.

The error ellipse semi-major (SMA) and semi-minor axes lengths along with the ellipse rotation angle reported by a LLS generally correspond to the characteristics of the 50% confidence ellipse, i.e., 50% of the located return strokes should have ground truth strike locations that occur within the error ellipse defined by the provided parameters. This error or confidence ellipse can in fact be calculated for any desired level other than 50% by scaling the semi-major and semi-minor axes of the 50% confidence ellipse according to Eq. (1).

$$SC = \frac{\sqrt{-2.\ln(1-P)}}{1.177} \quad (1)$$

with SC the resulting scaling factor belonging to the desired probability P. More details about the confidence ellipse can be found in Stansfield (1947), Cummins et al. (1998), and Diendorfer et al. (2014). In any case, an alternative way to look at the location quality is to monitor the SMA behavior. From Table 1, it follows that the SMA for BR is highest, indicating that the

location quality is lower compared to the other data sets. It also confirms the LA values retrieved by the method described above.

$\chi^2$ values provide additional insight about the accuracy of the error ellipse parameters. A standard distribution of the $\chi^2$ has a mean value of 1, whereby 1% of the $\chi^2$ values are larger than 5. It is expected that the distribution of the SMA of the 50%
confidence ellipse is close to the median location accuracy if all systematical errors are removed and random errors are based on the real measurement errors (Nag et al., 2015). For all the LLSs, except BR, the mean $\chi^2$ is about 1, with only a few percent of the strokes exhibiting a $\chi^2$ greater than 5 (ranging from 0.51%-2.21%). The mean $\chi^2$ value in BR is the largest at 4.11, with more than 20% of the values greater than 5. The latter suggests that many of the location errors in BR will be much larger, i.e., two to three times, than what is provided by the ellipse estimates.

Estimated (measured) median peak current values for 1[st] strokes, subsequent strokes, NGCs and PECs are also presented in Table 1. As expected, the 1[st] strokes exhibit larger absolute peak currents compared to the subsequent strokes, analogue to the peak current values of NGCs versus PECs. Since higher peak current strokes tend to be detected on average by a larger number of lightning sensors, the coordinates appointed by the LLS are likely to be of higher accuracy compared to strokes exhibiting a lower peak current. This may influence the probability of the algorithms to distinguish correctly between a new GSP or a
PEC, as will be discussed later on.

Finally, values for the mean and median separation distance between the first stroke in the flash, i.e., 1[st] GSP, and subsequent GSPs within the flash are illustrated in Table 1 as well. The position of the respective GSPs is calculated as the mean location of the strokes assigned to the GSP, whereby a weight is given inversely proportional to the respective semi-major axis of the stroke. The 99[th] percentiles are indicated together with the maximum estimated separation distance. In case this maximum is
found to be much larger than the 99[th] percentile, it indicates that the maximum is a one-off. Median values of the separation distance vary between 1.3 km (US) and 2.75 km (BR). The retrieved median distances agree well with those found in previous studies such as Stall et al. (2009) and Thottappilil et al. (1992). The maximum separation distance for AT, FR and SA is quite large, whereby for SA a few considerable separation distances are retrieved close to the maximum as evidenced by the value of the 99[th] percentile. It is essential to highlight that the large maximum separation distances could well be the result of a
location error by the LLS or a consequence of the manual grouping methodology based on the video information. From the perspective of cloud charge centers and the horizontal extent of downward leaders, it would make more sense to trace the lightning leader back to the location of the preliminary breakdown and only group strokes that emanate from a common charge region. However, this would require observations made by an LMA.

## 4. Algorithms

The sole purpose of a ground strike point algorithm is to group the different strokes of a flash into one or more ground strike points. The ultimate goal is to mimic, as accurately as possible, the exact distribution of GSPs compared to what is observed

in the high-speed camera images. The ability to do so enables the user to determine, with a high degree of certainty, on a predefined geographical and periodical scale, the ground strike point density based on a large set of actual LLS observations. To our knowledge, four such GSP algorithms exist to date. One of those has been described by Cummins et al. (2012). The

empirical formulae that resulted from that analysis was based on LLS data employing wave shape information from IMPACT sensors. Since in this study, the LLSs described in Section 2 utilize so-called LS700x sensor technology of Vaisala (except for BR), it is believed that this particular method is unsuitable to be applied directly to the data in this study and is hence disregarded (Cummins, private communication). In what follows, the three remaining algorithms are described.

## 4.1. Algorithm 1 (A1)

Developed by MTRG, this iterative K-means method works as follows. During the first iteration, the first stroke in the flash is taken as the location of the first GSP. Then subsequent strokes are assigned to a GSP if and only if the distance falls within a pre-defined minimum geometrical distance threshold. If the distance between the stroke and the previously determined GSPs is greater than this threshold, the stroke creates a new GSP, otherwise it is assigned to the closest GSP. Before an iteration ends, the GSP positions are updated according to the mean locations of the strokes assigned to the GSP, whereby a weight is

given to each stroke inversely proportional to the respective SMA, i.e., strokes with smaller SMAs will influence the GSP location more than strokes with large SMAs. Then a new iteration can start and the process is repeated until the mean GSP positions do not vary anymore; meaning all the strokes are durably assigned to their ground contact. It is important to mention that strokes with peak current $|I_p| < 6kA$ and/or with SMA values above 2 km are assigned to the previous GSP regardless of their position. For further details on this algorithm, the interested reader is referred to Pedeboy et al. (2012).

## 4.2. Algorithm 2 (A2)

This iterative K-means method has been developed and described in great detail in Campos et al. (2015, 2016). As a first step, strokes are sorted into two main groups, i.e., those with low and those with high SMA values based on a user-defined threshold. Initially, the algorithm tries to group the strokes with low SMAs among themselves, thereby creating the first set of GSPs. To do so, the mean location among the low-SMA strokes is first calculated. Then the algorithm checks the spherical distance between each low-SMA stroke and this mean location. The resulting distances are then compared against a threshold. This

threshold depends on the properties of the strokes in the flash, defined as twice the maximum SMA value among the low-SMA strokes in the flash. If all distances fall below the threshold, the low-SMA strokes are grouped within one GSP. However, in the case where one distance is larger and the rest are smaller than the threshold, a new iteration starts whereby now two potential GSP locations are tested. The first GSP adopts the location of the stroke in the previous iteration with the distance larger than the threshold, and the location of the second GSP is the mean of the locations of the other strokes. The algorithm

repeatedly checks the distances up to the point that the greatest distance between a GSP and all its associated strokes is smaller than the threshold, implying that the low-SMA strokes are grouped in a fixed set of GSPs. Subsequently, the algorithm attempts to group the strokes with high-SMA values into the previous retrieved GSPs, according to an elliptical scaling method

described in more detail in Campos et al. (2015). In order to do so, the error ellipse is scaled until it intersects with the location of one of the GSPs. The scaling value indicates how many times the scaled ellipse is larger or smaller than the original error ellipse. A maximum elliptical scaling factor of two is adopted in this study. If the scaling factor is below two, then it is assigned to that GSP and not otherwise. Finally, the algorithm groups redundant GSPs if the distances are smaller than the threshold used to split strokes into strokes with low- and high-SMAs.

### 4.3. Algorithm 3 (A3)

The most recent method has been introduced by Matsui et al. (2019). This non-iterative approach excels in its simplicity whereby a stroke with a distance below a certain threshold is assigned to an existing GSP when the 50% probability ellipse overlaps with one or more of the other error ellipses of strokes already assigned to that GSP. The GSP location is updated directly as the mean of the locations of the strokes. If not, a new GSP is created and the distances of the subsequent strokes are tested against the locations of the already existing GSPs produced by the algorithm.

Before going any further, it is appropriate to add following remark. The three algorithms described above somehow all rely to a certain degree on the availability of the strokes' SMA information at some point in the algorithm. However, not all existing LLSs provide details about the strokes' confidence ellipse. Especially in case of A3, this would mean that GSPs are determined solely by some prescribed separation distance and consequently coincides with A1.

### 4.4. Some initial examples

The flashes displayed here are examples of real flashes from the data set of this study and are specifically chosen to explain the principles employed by the algorithms in a clear manner. Of course, more complicated flashes exist with higher multiplicities.

Figure 1a displays a two-stroke flash with the original error ellipses displayed as solid lines. The peak currents of the strokes are -11.3 kA and -3.5 kA respectively. The strokes are about 850 m apart and have SMA values of 400 m and 1 km respectively. A1 will always group the strokes together in one GSP irrespective of their distance, since the second stroke has an absolute peak current smaller than 6 kA. For A2, adopting a distance threshold of, e.g., 500m results in stroke 1 being the first GSP as it is the only low-SMA stroke in the flash. Stroke 2 is in this case regarded as a high-SMA stroke and elliptical scaling is applied. The scaled error ellipse is displayed as the dotted ellipse in the plot. The error ellipse is scaled by a factor of less than two before it intersects with the location of the GSP. Therefore, these strokes will be grouped into one ground strike point. In the case of A3, the error ellipses overlap, therefore the grouping depends solely on the chosen distance threshold. If the threshold is below 850 m, then it will create two GSPs otherwise the strokes are grouped into a single GSP.

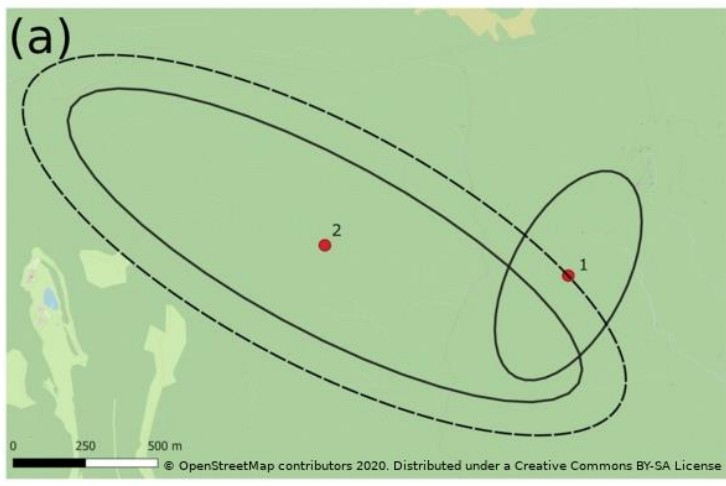

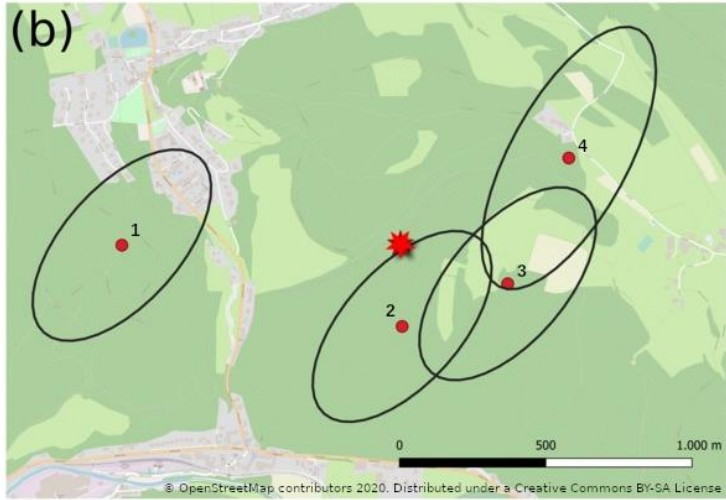

**Figure 1: a) Example of a two-stroke flash. The original error ellipses are displayed (solid) alongside the scaled error ellipse of stroke 2 (dashed) as used by A2. b) A flash with multiplicity 4. The star denotes the average position of all four strokes.**

The composition of the four strokes from another flash is visualized in Fig. 1b. The first three have an SMA of 400 m, while
250    the fourth stroke has an SMA of 500 m. If a distance threshold of 200 m is adopted, A1 will create four GSPs accordingly, since the distances are all larger than 200 m for all combinations possible. If the threshold is increased to 1 km however, the algorithm results in one GSP. For A2, let's take a threshold of 200 m to separate the low and high-SMA strokes. All strokes are considered as high-SMA events and therefore only elliptical scaling is applied. Since the error ellipses overlap already a lot, it is possible to envision that the scaling factors will be below two and therefore the last three strokes will be grouped into
255    a single GSP, while the 1st stroke is a GSP on its own. When adopting 1 km as threshold, all strokes are considered as low-

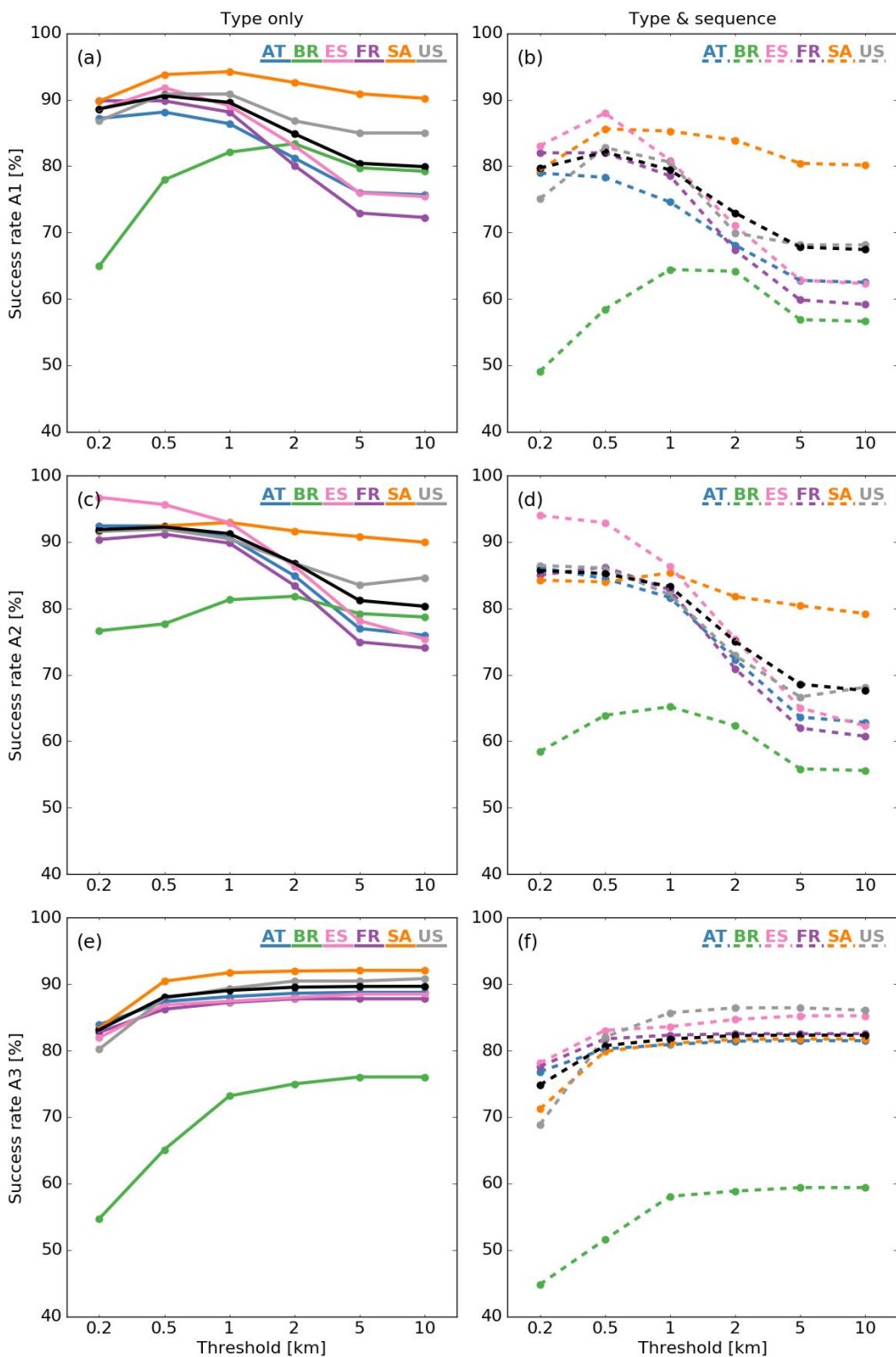

**Figure 2:** The success rate for the three algorithms is displayed in the "Type only" case (left plots) or the "Type and Sequence" (right plots) for algorithm A1 (a and b), A2 (c and d) and A3 (e and f). Colors are linked to each specific data set, whereas the black curve indicates the average result without Brazil.

SMA strokes and only spherical grouping is applied. First, the mean location of all four strokes is calculated, highlighted by the star in the plot. Then the distances of the strokes to the star are calculated. Since the distances are all below 1 km, they are grouped into a single GSP. In the case of A3, the first stroke will always be a GSP on its own since the error ellipse does not overlap with any of the other three. Depending on the adopted distance criterion, the algorithm results in either two or more GSPs.

## 5. Results

In the following analysis, a similar strategy is applied to all three algorithms. First of all, the ability to distinguish between a stroke creating a NGC and one that follows a PEC is examined. The latter will be denoted as the "Type only" criterion. Secondly, a stricter "Type and Sequence" criterion is validated. The latter not only checks whether the correct type is retrieved but additionally whether the order of occurrence is correct. By this it is meant that in the case of an NGC, whether it is correctly assigned as the $1^{st}$, $2^{nd}$, $3^{rd}$, ... GSP in the flash, while in the case of a PEC, if it is assigned to the correct GSP as retrieved from the video images.

A1 and A3 have one obvious threshold in common, i.e., the distance to group strokes into a particular GSP. In the case of A2, only the low-SMA strokes are grouped according to a flash dependent distance threshold. However, to facilitate the comparison of the three algorithms, the plots on the left and right in Figure 2 display the probability of the algorithms to correctly assign the "Type only" and "Type and Sequence", respectively, of the strokes as a function of the distance threshold ranging from 200 meters up to 10 kilometers. The latter threshold is exactly the distance threshold used by A1 and A3, while in the case of A2 it is the threshold that subdivides the strokes into low- and high-SMA strokes, followed by the algorithms' specific designed distance threshold.

As will be demonstrated later on, the trend for AT, FR, SA, ES and US is similar for each specific algorithm, while BR exhibits a different behavior. It is believed that this behavior of BR is a consequence of the low LA of the LLS observations at that time, prohibiting the algorithm to utilize its full potential. For this reason, the overall success rate of the algorithms, as denoted by the black curve in Fig. 2, is calculated without taking into account BR. Results hereof are quantified in Table 2.

### 5.1. Results A1

The success rate in determining the type (and sequence) of the strokes is plotted in Figure 2a (2b). Although only better by one or two percent, the best overall "Type only" success rate of 90.6% is found adopting a distance threshold of 500 m. The algorithm displays a similar behavior for the "Type and Sequence" criterion, with an overall best of 82.1% at 500 m. Overall, a 10% to 15% drop is noticed if the sequence is additionally taken into account as criterion. In Table 2 the results for "Type only" are split into the classification success for NGC or PEC. Increasing the distance threshold in the algorithm leads to strokes being grouped more and more into a single GSP. As such, strokes are gradually more frequently allocated as a PEC by the algorithm. This explains the success rate of almost 100% for PEC at the largest threshold of 10 km. Similar reasoning can explain the behavior of NGC, whereby NGC is better predicted than PEC at lower distance thresholds.

**Table 2: Performance results for the three algorithms excluding BR, i.e., black curve in Fig. 2. Values in parenthesis are success rates for events without first strokes, i.e., thereby removing all single stroke flashes as well as the first strokes in the multiple-stroke flashes.**

| Distance threshold [km] | 0.2 | 0.5 | 1 | 2 | 5 | 10 |
|---|---|---|---|---|---|---|
| **Algorithm 1** | | | | | | |
| Type only correct [%] | | | | | | |
| All strokes | 88.6 (82.4) | 90.6 (85.8) | 89.6 (84.3) | 84.9 (77.1) | 80.4 (70.3) | 79.9 (69.5) |
| NGC | 92.0 (79.0) | 89.7 (72.5) | 84.9 (58.9) | 74.6 (30.5) | 65.2 (4.4) | 63.8 (0.8) |
| PEC | 84.4 | 91.6 | 95.3 | 97.3 | 98.9 | 99.4 |
| Type & Seq correct [%] | | | | | | |
| All strokes | 79.7 (69.2) | 82.1 (72.9) | 79.4 (68.8) | 72.9 (58.8) | 67.9 (50.8) | 67.4 (50.3) |
| **Algorithm 2** | | | | | | |
| Type only correct [%] | | | | | | |
| All strokes | 91.9 (87.5) | 92.4 (88.0) | 91.2 (86.5) | 86.8 (79.7) | 81.2 (71.0) | 80.3 (69.6) |
| NGC | 95.3 (87.1) | 93.6 (82.3) | 89.9 (71.8) | 79.9 (44.0) | 67.2 (8.6) | 64.6 (1.3) |
| PEC | 87.6 | 90.6 | 92.9 | 95.2 | 98.1 | 99.4 |
| Type & Seq correct [%] | | | | | | |
| All strokes | 85.7 (78.0) | 85.2 (77.3) | 83.3 (74.2) | 75.0 (61.5) | 68.6 (51.6) | 67.6 (50.1) |
| **Algorithm 3** | | | | | | |
| Type only correct [%] | | | | | | |
| All strokes | 83.1 (74.0) | 88.1 (81.6) | 89.1 (83.1) | 89.6 (83.9) | 89.7 (84.1) | 89.7 (84.1) |
| NGC | 99.1 (97.5) | 98.3 (95.4) | 98.3 (95.2) | 98.3 (95.2) | 98.3 (95.2) | 98.3 (95.2) |
| PEC | 63.8 | 75.6 | 77.9 | 79.0 | 79.2 | 79.3 |
| Type & Seq correct [%] | | | | | | |
| All strokes | 74.8 (61.2) | 80.8 (70.3) | 81.8 (71.9) | 82.3 (72.7) | 82.3 (72.8) | 82.3 (72.8) |

First strokes in the flash, including single stroke flashes, are per definition always correctly assigned by the algorithm. Hence, neglecting first strokes, i.e., removing all single stroke flashes as well as the first stroke in the multiple-stroke flashes, results in a decrease of the success rate by about 5% to 10%. The latter is indicated by the results between brackets in Table 2. By doing so, the results are not biased by the percentage of single stroke flashes in the individual regions. On the other hand, neglecting first strokes does not affect the PEC classification.

The effect of not using the condition to group strokes with $|I_\mathrm{p}| < 6$kA and/or SMA $> 2$km in the previous GSP regardless of its location, results in a minor drop of the success rate by not more than 1 %. This is as expected since only a limited number of strokes fall within this category.

Finally, it is worth mentioning that for an algorithm depending on solely a distance criterion to group strokes into GSPs, the success rate in the limit of very low and very high distance thresholds can be determined theoretically. This is true since all strokes will create a new GSP using the algorithm at very low distance thresholds while at very high distance thresholds, all strokes are grouped into a single GSP. Making use of the observed number of flashes, strokes and GSPs, the success rate can then be determined at those boundary conditions. The average number of GSP per flash in the case of SA is lowest among the data sets, resulting in the best performance at high distance thresholds.

### 5.2. Results A2

The success rate in determining the type (and sequence) of the strokes is plotted in Figure 2c (2d). To reiterate, in the case of A2, the threshold displayed on the x-axis is the threshold that sorts strokes into low- and high-SMA strokes. As such, toward the left side of the plot some strokes will be regarded as large SMA strokes because the algorithm applies a combination of spherical grouping and elliptical scaling. On the other hand, at large distance thresholds, most of the strokes, if not all, are regarded as small SMA strokes and only spherical grouping is utilized. At a threshold of 10 km the outcome resembles the outcome of A1, due to the merging of the GSPs, if the distances are below 10 km. Hence, on this side of the plot most, if not all, flashes have one single GSP. At 200 m, the algorithm performs better for BR compared to the other two algorithms, a consequence of the elliptical scaling. In fact, the primary motivation behind implementing the hybrid scaling method used by this algorithm was to increase the performance in case of low sensor density networks or near borders. Hence, under the latter conditions the use of this algorithm is recommended. However, the success rate for BR remains low compared to the other data sets at low thresholds. Looking at Table 2, A2 performs best at the 500 m threshold with an overall "Type only" success rate being about 2 to 4 % higher than A1 and A3, respectively, and is similar in case of "Type and Sequence".

### 5.3. Results A3

Figures 2e and 2f plot the success rate of correctly assigning the "Type only" and "Type and Sequence" in the case of A3. Compared to the previous two algorithms, the behavior exhibits a different pattern whereby the outcome for all data sets increases gradually up to a distance threshold of about 1km, after which the curve flattens out. Additionally, what is striking is that the results for the data sets are close to each other all over the line within approximately 5%, except for BR. The reason

why an enlarged distance threshold has practically no effect beyond 1 km is the explicit condition that the 50% probability ellipse needs to overlap with one or more of the other error ellipses of strokes already assigned to the GSP. Hence, this prerequisite prevents grouping strokes located at large distances from each other into a single GSP, as opposed to A1, for example. One can conclude that for A3, the distance threshold dominates at thresholds smaller than the average SMA values observed in Table 1, whereas the SMA values quickly become more important at larger distance thresholds. This is true, since for large thresholds it can be assumed that all strokes are within the threshold distance. The decision to group these into a GSP is determined by whether the ellipses overlap or not. Similar to A1 and A2, the data for BR exhibits the worst probability of success.

The results for the black curve in Fig. 2e and 2f are quantified in Table 2. A somewhat smaller drop than 10% in the success rate is observed going from "Type only" to "Type and Sequence". A more detailed look at the classification success of NGC and PEC reveals that the behavior is different when compared to the other algorithms. Here, NGC's classification success is rather stable over the entire line. Moreover, excluding first strokes does not have such a dramatic effect on the outcome as opposed to A1 and A2.

## 5.4. Dependence on the estimated peak current

Figure 3 plots the overall performance of the algorithm to determine the type of the strokes as a function of the median absolute peak current $|I_p|$. The results are presented, adopting the threshold of 500 m for all three algorithms. A different symbol and color is used for each of the four possible combinations, with open symbols denoting the results when first strokes are neglected, i.e., thereby neglecting single stroke flashes as well as the first strokes in multiple-stroke flashes.

For all the algorithms, the correctly assigned NGC strokes (green triangles) have a median $|I_p|$ that is larger compared to the incorrectly assigned ones (red triangles). This difference is more pronounced in the case of A1 and A3, while it is only 1 kA for A2. The smaller difference in median $|I_p|$ between the correctly and incorrectly assigned NGC strokes in case of A2 indicates that the correct classification is less dependent on the stroke's peak current compared to the other two algorithms.

The effect of neglecting the first strokes (open symbols) has been discussed before. A drop is noticed in the success rate of the algorithms according to the results listed in Table 2 (open triangles). While for A1 and A3, a similar behavior is found in terms of the median peak currents, for A2 it is found that the absolute median $/I_p|$ for incorrectly assigned NGCs is slightly larger by 0.5 kA as compared to the correctly assigned ones.

Similarly, one can look at the peak currents of the PECs. In the case of A2 and A3, correctly assigned PECs (green squares) have larger absolute median $|I_p|$ compared to the incorrectly assigned ones (red squares), whereas the opposite is found for A1.

The performance of A1 related to PECs is a consequence of assigning strokes with an absolute peak current below 6 kA, and/or SMA value larger than 2 km, to the previous GSP regardless of its position. As such, those particular low peak current strokes reduce the median peak current of correctly assigned PEC strokes in Fig. 3a.

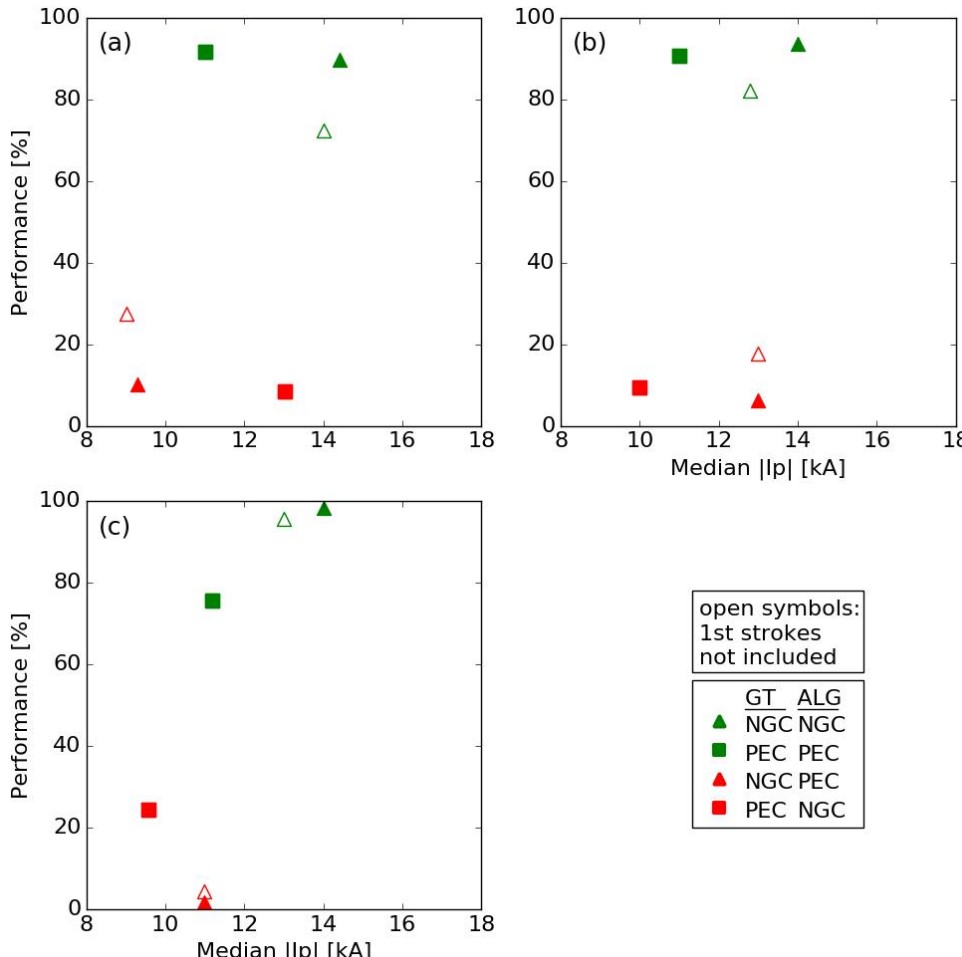

**Figure 3:** Algorithm performance as a function of median absolute peak current for (a) A1, (b) A2, and (c) A3. The threshold for which the results are presented is 500m for A1 and A2, and 10km for A3. The different symbols and colors denote the four possibilities, whereby a green (/red) color indicate that the algorithm (ALG) correctly (/incorrectly) assigned the type of the stroke compared to the ground-truth observations (GT). Ignoring first strokes in the flash results in the open symbols in the plots.

To conclude, it follows that, in general, larger absolute peak current strokes are more likely to be correctly classified as either
360  a NGC or a PEC. This is not surprising since larger absolute peak current strokes are on average reported by an increased number of lightning sensors, thereby locating the stroke more accurately.

## 6. Discussion and Conclusions

Three different ground strike point algorithms have been assessed in terms of their ability to correctly group strokes into ground
365  strike points. Since it is known that flashes tend to have more than one ground termination point on average, it is advisable to use ground strike point densities as opposed to flash densities derived from LLSs. The input for the algorithms is provided by

the observations made by local LLSs, whereas high-speed observations deliver the ground-truth observations against which the outcome of the algorithms is tested. Although some differences are noticeable among the algorithms, all three of them perform well with success rates up to 90% to retrieve the correct type of stroke in the flash. This means that in 90% of the cases, the number of ground strike points are retrieved as how they actually occurred in nature. Even though 100% is not reached, the use of GSP densities after applying a GSP algorithm to group the individual strokes within a flash in ground strike points will result in a significant improvement to assess the risk posed by lightning.

Note that the occurrence of forked strokes has been investigated in Poelman et al. (2021, nhess-2021-12, companion paper). However, the different ground strike points created by those forked strokes are inherently difficult to be disentangled by LLSs, especially when the forked contact points are close to each other. Hence, applying the algorithms described in this manuscript, it follows that it would result in an underestimation of the ground strike points.

It is further worth mentioning that the performance results of the different algorithms are biased by the specific flash multiplicity and ground strike point characteristics in the region. Furthermore, the quality of the local LLS is of particular importance in the success rate of the algorithm. Looking at the change in success rate depicted in Figure 2, one could conclude that adopting a distance threshold proportional to three to five times that of the mean LA results in the best success rate of the algorithms.

All three algorithms, with their proper characteristics, are high-performance tools both in speed and accuracy to group strokes into ground strike points. It is difficult to favor one algorithm over the other, whereas in absolute terms A2 performs the best, but only by a few percent. However, it is also the most complicated algorithm among the three, combining spherical grouping and elliptical scaling. The other two algorithms either solely depend on a distance and/or overlap of the error ellipses and are more straightforward to implement by the user.

**Code and data availability**

All codes and data processed could not be made available for public. For the access, the first author can be contacted by email: dieter.poelman@meteo.be

**Author Contributions**

DRP and WS conceptualized the research and carried out the analysis. SP, LZSC and MM provided the code of the algorithms. WS, SP, DH, MS, HH are strongly involved in the collection and preparation of the used data sets. DRP prepared the manuscript with review and editing from all co-authors.

**Competing interests**

The authors declare that they have no conflict of interest.

## Acknowledgments

The work done by the reviewers proved invaluable. Their critical reading and contribution of ideas and comments is very much appreciated. It has undoubtedly given more depth to the content and the way the results are presented. The authors would like to express their gratitude for the time and effort that has gone into it. The authors would like to thank the South African Weather Services (SAWS) for the use of the SALDN data, specifically M. Gijben, A. van der Merwe and M. Hartslief. Additional thanks go to T. Warner and C. Schumann for making available the South African high-speed video footage, to C. Mata for his 405 effort related to the USA data set and to X. Delorme, J. B. Varela, and A. R. Espana for their help in connection with the ground-truth data taken in France and Spain. Special thanks go to H. Kohlmann who assisted DRP in using QGis to correctly plot the error ellipses in appropriate projections used throughout this study. Finally, HH would like to thank the National Research Foundation of South Africa (Unique Grant No: 98244).

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
