# Peer review of "Global ground strike point characteristics in negative downward lightning flashes — part 2: Algorithm validation"

_Natural Hazards and Earth System Sciences, 2021_

## Referee Comment (RC1)

Review of nhess-2021-14 manuscript:

Global ground strike point characteristics in negative downward lightning flashes – part 2: Algorithm validation

Dieter R. Poelman et al.

This is well-written and very interesting work that is relevant to all areas associated with ground-based lightning risk and protection. This second paper in the sequence follows nicely after the first paper that describes cloud-to-ground (CG) ground strike-point (GSP) behavior in terms of "ground truth" using high-speed video observations in many different locations. The specific questions in this second contribution are "how well can existing LLS networks capture the strike-point behavior, and is there a 'best' algorithm for doing this?".

This work is a real contribution to these questions, but there is room for improvement, as indicated below. Some of these issues are important concerns that the authors need to address before I can recommend this work for publication. Some further suggestions/comments would be easy to address and would improve the contribution. Others may require additional effort that the authors may see as "future work." This is followed by a few technical corrections and suggestions.

Concerns

1.  An important issue with multiple ground contacts is the closely-spaced strike locations resulting from forked lightning and a fraction of the sequential strokes. It seems that the authors may have the information available to specifically determine the fraction of such strike points that at not properly classified by each LLS in the study, possibly as a function of distance projected into the plane of the camera. It may be that a detailed assessment of this is beyond the reasonable scope of this current work, but some discussion of this issue would be important to include, at least in the discussion section.

2.  All three GSP algorithms rely heavily on the error ellipse parameters, so the accuracy of these parameters is essential. The rather large disagreement between the measured LA and estimated SMA values in Table 1, at least for the BR LLS, call this into question. Helpful insight about the accuracy of the ellipse parameters for each LLS can be provided by the distribution of Ch-square values. Even just knowing the mean and standard deviation of the Chi-square values for each LLS (for the flashes included in the study) would be helpful.

3.  The phrase "semi-major axis" is used on line 73, with no earlier definition or explanation. Since Location error estimates are central to this work, the authors need to

precede this with a discussion of what an error ellipse is and what an SMA is, either in a short section of its own or through references to earlier work. A number of LLS's do not have these parameters, and this should probably be mentioned. In those cases, GSPs would probably be determined solely by some prescribed separation distance.

4. The findings in both papers indicate that ground strike point statistics differ in different regions. This is worth mentioning.

5. All the regions exhibited higher percentages of single stroke flashes than any of the seven studies cited in the 2013 CIGRE Brochure 549 (Table 2.1), which is relevant but not cited. The authors should address this apparent discrepancy.

Suggestions/Comments/Questions

6. Have the authors collected information about stroke order, SMA, separation distance to closest GSP, or nearest-sensor risetimes for the mis-classified strokes? As noted by Stall et al. (2009) and Cummins (2012), stroke order and risetime information might be helpful in resolving strokes (PEC/NGC) that create new ground contacts. Maybe this is future work?

7. Is there an explanation for the unusual GSP statistics for SA (sorry if I missed it) (maybe distance-to-camera?) Are flashes that interacted with towers in the SA dataset excluded from the analysis?

8. Is there an explanation for the uniformly-low peak current for the ES LLS, for all stroke types? This does not need to go into the manuscript, unless the authors find it particularly relevant.

9. The distance information in Table 1 is a "treasure trove" of observations that are not explored in this work. First, the very large maximum distances for most LLS's are much larger than what was reported in the past, which is at-odds with the authors' statement on lines 152-153. Maybe these are caused by individual location errors? Do these values match with the video, and with the logical definition of a flash? Also, knowing the fraction of separation distance near the spatial accuracy of the LLS data (below 500m) would be interesting and helpful information. Maybe this is future work?

Technical Corrections/Suggestions

10. Throughout the manuscript, the term "amount" is used when talking about the number of strokes or sensors. The same is true in paper 1. English usage typically employs "number" to quantify integer values like these.
11. Line 33: suggest "should not" rather than "cannot"
12. Line 35: suggest "nature and society" rather than "human safety"
13. Line 46: suggest adding Koshak et al. (2015) along with the existing Yair (2018) reference
14. Line 57: suggest changing "…relate Ng to…" to "…infer Ng from…"
15. Line 59: suggest changing "…flash is…" to "…flash has historically been…"
16. Line 87:  the acronym "CC" is used before it is defined
17. Line 102: suggest adding "flash" before "detection efficiencies"
18. Line 104: should clarify what is meant be a "downward event"
19. Line 111: suggest adding Zhu et al. (2020) as a good reference for tower-based LA estimates
20. Line 126: the phrase "one can find" leaves the reader hanging. It seems like this should refer back to the references in section 2
21. Line 127: remove "can" ?
22. Line 128: suggest adding "and are described in the text that follows" after "data set"
23. Line 132: suggest replacing "LA" with "random location errors", since this method does not quantify local mean (bias) errors.
24. Line 160: suggest adding "waveshape information derived from" before "IMPACT"
25. Line 162: suggest adding "directly" after "applied"
26. Lines 182-184: The use of "so-called spherical threshold" makes the algorithm seem rather mysterious. I think that it behaves simply as an Euclidian distance at this point in the algorithm.
27.  Line 200: should clarify what is meant by "pre-existing GSPs."

Suggested References

Koshak, W.J., Cummins K.L., Beuchler D.E., and other (2015), Variability of CONUS Lightning in 2003–12 and Associated Impacts, j. Appl. Met. & Clim., v54, pp 15-41, dio: 10.1175/JAMC-D-14-0072.1, 2015.

Zhu, Y., Lyu, W., Cramer, J., Rakov, V., Bitzer, P., & Ding, Z. (2020). Analysis of location errors of the U.S. National Lightning Detection Network using lightning strikes to towers. Journal of Geophysical Research: Atmospheres, 125, e2020JD032530. https://doi.org/ 10.1029/2020JD032530

Kenneth Cummins, March 2021

---

## Author Comment (AC1)

**Answer to RC1 (Ken Cummins), nhess-2021-13**

**General comments**

This is well-written and very interesting work that is relevant to all areas associated with ground-based lightning risk and protection. This second paper in the sequence follows nicely after the first paper that describes cloud-to-ground (CG) ground strike-point (GSP) behavior in terms of "ground truth" using high-speed video observations in many different locations. The specific questions in this second contribution are "how well can existing LLS networks capture the strike-point behavior, and is there a 'best' algorithm for doing this?". This work is a real contribution to these questions, but there is room for improvement, as indicated below. Some of these issues are important concerns that the authors need to address before I can recommend this work for publication. Some further suggestions/comments would be easy to address and would improve the contribution. Others may require additional effort that the authors may see as "future work." This is followed by a few technical corrections and suggestions.

**Concerns**

1. An important issue with multiple ground contacts is the closely-spaced strike locations resulting from forked lightning and a fraction of the sequential strokes. It seems that the authors may have the information available to specifically determine the fraction of such strike points that are not properly classified by each LLS in the study, possibly as a function of distance projected into the plane of the camera. It may be that a detailed assessment of this is beyond the reasonable scope of this current work, but some discussion of this issue would be important to include, at least in the discussion section.

**=> The occurrence of forked strokes will be highlighted in the new version of the companion paper (nhess-2021-12) following the suggestion posed by nhess-2021-12-RC2. Forked stroke statistics are retrieved for AT, BR, SA, & US.**

|  | Flashes containing ≥1 forked stroke(s) | %forked strokes in those flashes | %forked strokes in overall data set |
|---|---|---|---|
| AT (2018 only) | 9.4% (18/191) | 34.4% (21/61) | 3.75% (21/560) |
| BR | 10.7 (13/122) | 21.8% (14/64) | 2.3% (14/619) |
| SA | 7.0% (34/484) | 20.8% (41/197) | 2.2% (41/1839) |
| US | 10.3% (8/78) | 42.8% (9/31) | 2.95% (9/305) |
| ALL | 8.3% (73/875) | 24.1% (85/353) | 2.55% (85/3323) |

**If one adopts 2-4% of the observed strokes being forked, this results in an increase of the average amount of ground strike points per flash, N(GSP/flash), as indicated in Table 1 of nhess-2021-12 by this same factor.**

**=> The different ground strike points created by forked strokes are inherently difficult to be disentangled by LLSs, especially when the forked contact points are close to each other. Hence, applying the algorithms described in this manuscript, it follows that it would result**

in an underestimation of the ground strike points. However, the distances between ground strike points of a forked stroke are normally smaller compared to the spatial accuracy of the LLS and are therefore not so important for this analysis.

**This will be included in the new version of the manuscript.**

2. All three GSP algorithms rely heavily on the error ellipse parameters, so the accuracy of these parameters is essential. The rather large disagreement between the measured LA and estimated SMA values in Table 1, at least for the BR LLS, call this into question. Helpful insight about the accuracy of the ellipse parameters for each LLS can be provided by the distribution of Chi-square values. Even just knowing the mean and standard deviation of the Chi-square values for each LLS (for the flashes included in the study) would be helpful.

**=> Indeed, $\chi^2$ values provide additional insight about the accuracy of the error ellipse parameters. A standard distribution of the $\chi^2$ with all the systematical errors corrected and the real standard deviations of the measurements applied has an expected mean value of 1, whereby maximum 1% of the $\chi^2$ values are larger than 5. With this in mind, the following is found for the different data sets:**

| | $\chi^2$ | |
|---|---|---|
| **LLS** | **Mean** | **% >5** |
| **AT** | 1.01 | 0.87 |
| **BR** | 4.11 | 21.88 |
| **FR** | 1.35 | 1.01 |
| **SA** | 0.67 | 0.51 |
| **ES** | 1.07 | 0 |
| **US** | 1.23 | 2.21 |

**It is expected that the distribution of SMA of the 50% confidence ellipse is close to the median location accuracy if all systematical errors are removed and random errors are based on the real error measurement errors\*. For all the LLSs, except BR, the mean $\chi^2$ is about 1, with only a few percent of the strokes exhibiting a $\chi^2$ greater than 5 (ranging from 0.51%-2.21%). Mean $\chi^2$ value in BR is the largest at 4.11, with more than 20% of the values greater than 5. One could conclude therefore that the LLS in BR at the time of observation was not correctly calibrated. This is also the reason why the success rates of the different algorithms are the lowest for BR in this study.**

**\*Nag, A. Murphy, M. J., Schulz, W., & Cummins, K. L. (2015). Lightning location systems: insights on characteristics and validation techniques. Earth and Space Science, 2(4), 65-93, https://doi.org/10.1002/2014ea000051.**

**This will be included in the new version of the manuscript.**

3. The phrase "semi-major axis" is used on line 73, with no earlier definition or explanation. Since Location error estimates are central to this work, the authors need to precede this with a discussion of what an error ellipse is and what an SMA is, either in a short section of its own or through

references to earlier work. A number of LLS's do not have these parameters, and this should probably be mentioned. In those cases, GSPs would probably be determined solely by some prescribed separation distance.

=> **Following will be added to the text before the term "semi-major axis" is used: "The error ellipse semi-major (SMA) and semi-minor axes lengths along with the ellipse rotation angle reported by a LLS generally correspond to the characteristics of the 50% confidence ellipse, i.e., 50% of the located return strokes should have ground truth strike locations that occur within the error ellipse defined by the provided parameters. Note that this error or confidence ellipse can in fact be calculated for any desired level other than 50% by scaling the semi-major and semi-minor axes of the 50% confidence ellipse according to Eq. (1)**

$$SC = \frac{\sqrt{-2.\ln(1-P)}}{1.177} \ (1)$$

**with SC the resulting scaling factor belonging to the desired probability P. More details about the confidence ellipse can be found in Stansfield (1947)[1], Cummins et al. (1998)[2], Diendorfer et al. (2014)[3]."**

=> **Following paragraph will be added to the discussion section: "Note that not all LLSs provide details about the strokes' confidence ellipse. In that case, GSPs can probably be determined solely by some prescribed separation distance."**

**[1] R. G. Stansfield, "Statistical Theory of D.F. Fixing," Journal of the Institution of Electrical Engineers-Part IIIA: Radiocommunication, vol. 94, no. 15, pp. 762–770, 1947.**

**[2] K. L. Cummins, M. J. Murphy, E. A. Bardo, W. L. Hiscox, R. B. Pyle, and A. E. Pifer, "A Combined TOA/MDF Technology Upgrade of the U.S. National Lightning Detection Network," Journal of Geophysical Research: Atmospheres, vol. 103, no. D8, pp. 9035–9044, Jan. 1998.**

**[3] G. Diendorfer, H. Pichler, W. Schulz, EUCLID Located Strokes to the Gaisberg Tower – Accuracy of Location and its assigned Confidence Ellipse. ILDC 2014**

4. The findings in both papers indicate that ground strike point statistics differ in different regions. This is worth mentioning.

**The authors propose to highlight the difference in the companion paper nhess-2021-12. In a new version of this manuscript (nhess-2021-13), a reference could be included to nhess-2021-12 concerning the different statistics and a sentence added in the discussion section.**

=> **in nhess-2021-12: It is found that the mean amount of ground strike points per flash is 1.56, varying in the four different regions from 1.29 to 1.90. While the maximum number of GSPs per flash just varies between 4 and 5, the mean number of strokes per GSP varies from 1.82 to 2.94.**

=> in nhess-2021-13: the 1$^{st}$ § in Sect. 6 'Conclusions' will be rewritten as follows: "… This means that in 90% of the cases, the number of ground strike points are retrieved as how they actually occurred in nature. It is worth mentioning that the performance results of the different algorithms are biased by the specific ground strike point characteristics in the region."

5. All the regions exhibited higher percentages of single stroke flashes than any of the seven studies cited in the 2013 CIGRE Brochure 549 (Table 2.1), which is relevant but not cited. The authors should address this apparent discrepancy.

=> **The percentage of single-stroke flashes as described in the 2013 CIGRE Brochure 549 (Table 2.1) range from 13% (New Mexico; Kitagawa et al., 1962) up to 21% (Sri Lanka; Cooray and Jayaratne, 1994). The values found in manuscript nhess-2021-12 (Table 1) are indeed somewhat higher ranging from 23% (BR) up to 38.4% (SA).**

**Note that in our study flashes are removed from the original set of flashes if at least one channel is partly visible, diffuse or simply out of the field of view. As such only flashes are kept of which the different GSPs could be determined with great confidence. The number of flashes that are removed from the data sets is minimal w.r.t. total number of flashes per data set. Anyhow, taking those removed flashes into account to re-calculate the percentage of single-stroke flashes it turns out that the percentage in case of AT drops from 29.2% to 28.2%, for BT from 23% to 21.4%, for SA from 38.4% to 37.9%, whereas it remains unchanged for US at 25.6%.**

**For AT: We would like to draw your attention to the recently published article by Schwalt et al. (2021)\* in which the authors investigate specifically the percentage of single-stroke flashes in Austria. It is found that the percentage of single-stroke flashes among all negative flashes is 27%. A possible dependency of the occurrence of single-stroke flashes with the underlying terrain (Alpine versus pre-Alpine) is found in this study. The 28.2% found in the present study is therefore in line with the findings of Schwalt et al. (2021).**

**For BR: the newly calculated percentage of 21.4% is only slightly higher compared to the 17% quoted in the 2013 CIGRE brochure 549.**

**For SA: Looking into the LLS data (thus not just the correlated high-speed camera cases) in a corresponding area as in this study and averaged over a few years, a similar value of 1.2 strokes per flash is found. It seems that this area, at an altitude of about 1600 m asl, is prone to single-stroke flashes. The origin of this discrepancy, compared to the other regions, is indeed worth a thorough investigation, but out of the scope of this particular study.**

**For US: We would like to draw your attention to Fleenor et al. (2008)\*\*. In this study 40% (41/103) of the negative cloud-to-ground flashes are single-stroke flashes. It was noted that the time-resolution of the camera was limited to 16.7 ms, which could lead to an underestimation of the true negative multiplicity by about 11% (Biagi et al., 2007). However, even taken this underestimation into account, the percentage of the single-stroke flashes in the present study is still in line with Fleenor et al. (2008).**

***Schwalt, L., Pack, S., Schulz, W., and Pistotnik, G. (2021). Percentage of single-stroke flashes related to different thunderstorm types, Electric Power System Research, 194, 107109

****Fleenor, S. A., Biagi, C. J., Cummins, K. L., and Krider, E. P. (2008). Characteristics of cloud-to-ground lightning in warm season thunderstorms in the Great Plains, 20th International Lightning Detection Conference, 21-23 April, Tucson, Arizona, USA.

=> We would like to mention that Table 2 of this manuscript includes the success rates of the different algorithms for all flashes (single and multiple-stroke flashes) as well as in brackets the results when first strokes are removed (thereby removing all single stroke flashes as well as the first stroke in the multiple-stroke flashes) so that the results are not biased by the percentage of single stroke flashes in the individual regions.

**Suggestions/Comments/Questions**

6. Have the authors collected information about stroke order, SMA, separation distance to closest GSP, or nearest-sensor risetimes for the mis-classified strokes? As noted by Stall et al. (2009) and Cummins (2012), stroke order and risetime information might be helpful in resolving strokes (PEC/NGC) that create new ground contacts. Maybe this is future work?

=> We agree that all the information is available to tackle the above suggestion. However, it is not directly the intention of this paper to improve upon the current described algorithms in the manuscript, but merely to examine the performance at this stage. We feel it to be more appropriate to investigate the above suggestion in more detail in a follow-up/future study.

=> The distribution of the percentage of subsequent strokes creating a new GSP as a function of stroke order has been examined in Fig. 5 of the companion manuscript nhess-2021-12. Indeed, the second stroke in the flash is most likely to produce a new ground termination, in analogy of the results presented in Stall et al. (2009).

=> Regarding the rise times, one should be cautious since rise times obtained using different type of LLSs may not be comparable to each other. Anyhow, in analogy with Stall et al. (2009), the median rise times for (1st+NGC) and for PEC respectively are 6.5μs and 6.5μs (AT), 4.8μs and 3.6μs (BR), 3.0μs and 1.6μs (SA), and 4.1μs and 3.85μs (ES). Thus a slightly larger rise time is found for 1st+NGC strokes compared to PECs in all the data sets except AT.

=> The mean SMA for 1st+NGC is about a factor of two smaller compared to the SMA of PECs, while the median values are the same for all data sets combined.

7. Is there an explanation for the unusual GSP statistics for SA (sorry if I missed it) (maybe distance-to-camera?) Are flashes that interacted with towers in the SA dataset excluded from the analysis?

**=> The original SA data set to start with contained both downward and upward events. The latter are indeed events triggered by the two tall towers located in Johannesburg – the Sentech and Hillbrow tower, approximately 250m each. However, for this study all tower events in the SA data set are excluded.**

**The average number of GSP per flash, N(GSP/flash), of 1.29 is the lowest among the data sets, as indicated in Table 1 of nhess-2021-12. The comments about SA written in point 5 are valid as well here.**

8. Is there an explanation for the uniformly-low peak current for the ES LLS, for all stroke types? This does not need to go into the manuscript, unless the authors find it particularly relevant.

**=> There seems to be no apparent reason for the low estimated median peak currents in ES. The same LLS of Météorage has been used for ES and FR. Anyhow, the peak current behavior, although minimal, among the different types of strokes is as expected.**

**=> The peak currents in ES have been checked against the peak current estimates of the EUCLID network. Note that at the time of the ground-truth observations, the LLS network of MTRG benefited from an additional set of LS7002 sensors employed in Spain which were not part of EUCLID. The differences in the peak current estimates between MTRG and EUCLID are minimal, and a similar uniformly-low peak current is found by EUCLID.**

9. The distance information in Table 1 is a "treasure trove" of observations that are not explored in this work. First, the very large maximum distances for most LLS's are much larger than what was reported in the past, which is at-odds with the authors' statement on lines 152-153. Maybe these are caused by individual location errors? Do these values match with the video, and with the logical definition of a flash? Also, knowing the fraction of separation distance near the spatial accuracy of the LLS data (below 500m) would be interesting and helpful information. Maybe this is future work?

**=> Point well taken. A closer look at the calculation of the GSP distances w.r.t. 1$^{st}$ ground strike point revealed some small bugs which have been taken care of. The biggest difference is now for BR, which previously contained 1 separation distance of ~150km(!). This particular (2-stroke) flash has now been excluded. Below the updated values for the distance between GSPs and the 1$^{st}$ stroke in the flash. The position of the GSP is calculated as the mean location of the strokes assigned to the GSP, whereby a weight is given inversely proportional to the respective semi-major axis of the stroke. Note that now instead of the 95$^{th}$ percentile, the 99$^{th}$ percentiles are indicated. In case the maximum separation distance is found to be much larger than the 99$^{th}$ percentile, it indicates more clearly than in previous version that the maximum is a one-off.**

|  | LLS | | | | | |
|---|---|---|---|---|---|---|
|  | **AT** | **BR** | **FR** | **SA** | **ES** | **US** |
| **Sample size** | 334 | 79 | 231 | 116 | 45 | 41 |
| **Mean (km)** | 2.42 | 3.03 | 2.43 | 3.73 | 2.84 | 1.48 |
| **Median (km)** | 2.05 | 2.75 | 2.19 | 2.27 | 2.51 | 1.30 |
| **99th percentile (km)** | 9.52 | 7.62 | 7.21 | 20.59 | 6.34 | 4.8 |
| **Maximum (km)** | 16.5 | 8.09 | 13.69 | 20.9 | 6.75 | 5.43 |

**=> The comment made at L152-153 of the original manuscript is related to the median value of 2.1 km found in Stall et al. (2009) and the quoted geometric mean (GM) of 1.7km in Thottappillil et al. (1992). For your interest, the GM in the data sets is 1.8 km (AT), 2.67 km (BR), 1.92 km (FR), 2.26 km (SA), 2.15 km (ES), and 1.04 km (US).**

**=> The maximum values found in this work are for AT, FR and SA a factor of about 1.5-2 larger than what is commonly used in the artificial definition of a flash, i.e., grouping strokes within a temporal window of one second of the first stroke, a maximum interstroke interval of 0.5 s and a maximum distance to the first stroke of 10 km. The few cases with a large separation distance have been rechecked. Imagery of those can be found below:**

- **SA, F456: 3 GSPs in order of appearance from left to right. Observation time GSP1 @ 30 s 359 ms followed by CC of 500 ms, GSP2 @ 31 s 41 ms, GSP3 @ 31 s 134 ms. $\chi^2$(GSP1) is 2, $\chi^2$(GSP2) is 1.6 and $\chi^2$(GSP3) is 0.5. The distance between GSP1 & GSP2 is 20.8, and GSP1 & GSP3 is 10.9 km. The large separation distance could be related to the LA error of GSP1 and GSP2 with $\chi^2 > 1$. If not, it is a result of the manual grouping methodology based on the video information.**

[Figure]

[Figure]

[Figure]

- **SA, F146: 2 GSPs in order of appearance from left to right. Observation time GSP1 @ 289 ms and GSP2 @ 476 ms. $\chi^2$(GSP1) is 4.1 and $\chi^2$(GSP2) is 0.1. The distance between the 2 GSPs is 13.5 km**

[Figure]

[Figure]

- **SA, F491: 7 stroke flash with 2 GSPs. Observation time GSP1 @ 562 ms, followed by subsequent stroke @ 593 ms with a CC of 476 ms. GSP2 @ 796 ms. $\chi^2$(GSP1) is 0.2 and $\chi^2$(GSP2) is 1.1. The distance between the two GSPs is 20.9 km. The large separation distance could be related to the LA error of GSP2 with $\chi^2>1$. If not, it is a result of the manual grouping methodology based on the video information.**

[Figure]

- **SA, F86: 6-stroke flash with 3 GSPs (GSPs shown from left to right in order of appearance). GSP1 observed @ 561 ms, GSP2 @ 654 ms, and GSP3 @ 744 ms. $\chi^2$(GSP1) is 0.3, $\chi^2$(GSP2) is 0.9, and $\chi^2$(GSP3) is 2.2. Distance between GSP1 and GSP2 is 19.3 km.**

[Figure]

- **SA, F215: 10 stroke flash with 5 different GSPs (from left to right, top to bottom). GSP1 observed @ 61 ms, GSP2 @ 104 ms, GSP3 @ 320 ms, GSP4 @ 380 ms, and GSP5 @ 442 ms. $\chi^2$(GSP1) is 2.9 and $\chi^2$(GSP2) is 0.2, $\chi^2$(GSP3) is 0.2, $\chi^2$(GSP4) is 2.7, and $\chi^2$(GSP5) is 0.3. The distance between GSP1-GSP3 is 15.2 km, distance GSP1-GSP4 is 14.1 km and distance GSP1-GSP5 is 17.5 km.**

- **AT, F0154: 2 stroke flash with 2 GSPs. Observation time GSP1 @ 736 ms, and GSP2 @ 849 ms. $\chi^2$(GSP1) is 1.9 and $\chi^2$(GSP2) is 2.3. Only 3 sensors participated in the location of the second stroke. The distance between the 2 GSPs is 16.5 km. The large separation distance could be related to the LA error of GSP2 with $\chi^2 > 1$. If not, it is a result of the manual grouping methodology based on the video information.**

[Figure]

[Figure]

- **AT, F0098: This was a flash having the maximum separation distance for AT of 25.4 km in the original manuscript. Examining somewhat in greater detail this particular flash it is found that the stroke assignment wasn't done properly. It turns out that the separation distances are now 2 and 3.5 km within this flash.**

**=> Except for BR (with bad LA) we expect the separation distance near the spatial accuracy of the LLS does not play a big role. This is true since looking at Fig. 2 of the manuscript, the drop in success rate between 500 m and 200 m is not that large. Only for the Algorithm 3 there is a drop of about 10%.**

**Technical Corrections/Suggestions**

**=> All the corrections and suggestions (number 10 to 27) will be taken into account in an updated version of the manuscript.**

10. Throughout the manuscript, the term "amount" is used when talking about the number of strokes or sensors. The same is true in paper 1. English usage typically employs "number" to quantify integer values like these.

11. Line 33: suggest "should not" rather than "cannot"

12. Line 35: suggest "nature and society" rather than "human safety"

13. Line 46: suggest adding Koshak et al. (2015) along with the existing Yair (2018) reference

14. Line 57: suggest changing "…relate Ng to…" to "…infer Ng from…"

15. Line 59: suggest changing "…flash is…" to "…flash has historically been…"

16. Line 87: the acronym "CC" is used before it is defined

17. Line 102: suggest adding "flash" before "detection efficiencies"

18. Line 104: should clarify what is meant be a "downward event"

19. Line 111: suggest adding Zhu et al. (2020) as a good reference for tower-based LA estimates

20. Line 126: the phrase "one can find" leaves the reader hanging. It seems like this should refer back to the references in section 2

21. Line 127: remove "can" ?

22. Line 128: suggest adding "and are described in the text that follows" after "data set"

23. Line 132: suggest replacing "LA" with "random location errors", since this method does not quantify local mean (bias) errors.

24. Line 160: suggest adding "waveshape information derived from" before "IMPACT"

25. Line 162: suggest adding "directly" after "applied"

26. Lines 182-184: The use of "so-called spherical threshold" makes the algorithm seem rather mysterious. I think that it behaves simply as an Euclidian distance at this point in the algorithm.

27. Line 200: should clarify what is meant by "pre-existing GSPs."

**Suggested References**

Koshak, W.J., Cummins K.L., Beuchler D.E., and other (2015), Variability of CONUS Lightning in 2003–12 and Associated Impacts, j. Appl. Met. & Clim., v54, pp 15-41, dio: 10.1175/JAMC-D-14- 0072.1, 2015.

Zhu, Y., Lyu, W., Cramer, J., Rakov, V., Bitzer, P., & Ding, Z. (2020). Analysis of location errors of the U.S. National Lightning Detection Network using lightning strikes to towers. Journal of Geophysical Research: Atmospheres, 125, e2020JD032530. https://doi.org/ 10.1029/2020JD032530

---

## Author Comment (AC2)

**General comments:**

I read this paper with great interest. This paper test the validity of three strokes-to-ground-strike-points grouping algorithms by comparing the outputs of the grouping algorithms with the ground-truth data derived from high-speed video observations in several regions. As I said in my comments for part 1 (the companion paper), this validation study can help current LLSs to derive strike points data from their existing data and such product is going to be a very important parameter for lightning protection/lightning risk assessment. See additional specific technical comments below.

**Specific Comments:**

1. Since SMA is not available for all LLSs, I recommend authors add A4 (even simpler than A3), which uses distance threshold only to determine if a stroke is PEC or NGC and show the results of the simplest method as well, just for comparison.

**=> The algorithm A4 as proposed by the reviewer has been indeed investigated by the authors. The reason for not including it in the manuscript is because its output does not vary much from the output of A1. This is not so surprising, since A1 decides between assigning a stroke as NGC or PEC based on a simple distance threshold. Only the SMA comes into play in A1 for calculating the GSP position as the weighted mean of the stroke positions in the GSP.**

**=> We would therefore opt to include a comment concerning "A4" and its output, rather than including it as an extra algorithm.**

2. In this paper, the authors listed the performance of three algorithms using different distance thresholds. In order to ensure a high success rate, the selection of such threshold in all algorithms is strongly dependent on the location accuracy of the network. I think that might need more discussions. Hopefully, authors can come up with a general guidance/rule regarding how to select the optimal distance threshold with respect to the location accuracy (or even more parameters) of the network.

**=> Looking at the change in success rate in Figure 2 for the different algorithms, one could conclude that adopting a distance threshold proportional to 3 to 5 times that of the mean LA results in the best success rate.**

3. Can authors share any insights on why larger peak current are more likely to be correctly classified (PEC vs. NGC)? Possibly related to location accuracy's dependence on peak current? Do large peak current CGs always have better location accuracy?

**=> The mean SMA for 1$^{st}$ strokes & subsequent NGC in a flash is 0.26 km, whereas it is 0.45 km for PECs for all data sets combined.**

**=> Combining the LLS information from the different data sets in this study it is found that the larger the absolute peak current of the stroke, the smaller the SMA is on average. Larger peak current strokes are reported by an increased number of sensors on average. The more sensors participate in a solution; the better the location accuracy will be of that particular stroke.**

4. Table 1, it would be nice to also give years when the ground-truth datasets were recorded.

**=> This will be included in a next version of the manuscript.**

5. Line 116, I think here you are referring to "electric field change sensor/meter", or fast/slow antenna. Field mill is usually referred to as the electrostatic flux meter that monitoring electric field intensity at ground over a long period but with time resolution usually in one second.

**=> Correct, this will be corrected in a next version of the manuscript.**

6. My understanding is that in your ground-truth dataset, you only kept flashes with at least two return strokes detected by the LLS. Correct?

**=> The ground-truth data sets include multiple stroke flashes as well as single-stroke flashes. However, since first strokes are (per definition) always correctly assigned by the different ground strike point algorithms, we have included in Table 2 the results when first strokes are excluded, i.e., results within brackets, and in Figure 3 we show results without first strokes as open symbols.**

7. Line 172, "repeated until the mean GSP positions do not vary anymore" I thought it is repeated till the last return stroke was assigned.

**=> The algorithm really ends when the GSP locations are stable, meaning no variation with the previous iteration, as has been described as such in the manuscript.**

8. In this study, there is no flash grouping (group strokes into flashes) on the LLS end involved in this study because a flash was first defined by the high-speed video data and LLS data were searched for the flash. Is my understanding correct?

**=> This is correct. The flash grouping is based on the video images. LLS data is then used to assign the location and other parameters belonging to the observed strokes. This will be stressed in a new version of the manuscript.**

9. Line 190, please provide a reference for the scaling method

**=> A reference is included at the beginning Section 4.2. "Algorithm 2 (A2)" to Campos et al. (2015, 2016). But it is indeed a good idea to reference once more to Campos et al. (2015) in which the ellipse scaling factor is clarified in more detail.**

Line 110, Here I am providing two additional references on CG validation studies of NLDN using videos published in JGR, with titles: 1) Upward lightning observations from towers in Rapid City, South Dakota and comparison with National Lightning Detection Network data, 2004-2010. 2) A study of national lightning detection network responses to natural lightning based on ground truth data acquired at LOG with emphasis on cloud discharge activity.

**=> Those particular references will be included in a new version of the manuscript.**

**Minor editorial suggestions:**

2. Line 59: "By definition, the location of a flash is determined by that of the first stroke in the flash." This is probably true for some of the LLSs (like NLDN or EUCLID). Some use centroid.

**=> Correct, a comment to this issue will be included in an updated version of the manuscript.**

3. Line 87, What are "CC discharges"?

**=> Typo. Intracloud (IC) discharges are meant.**

4. Line 61, It is not clear to me what does subscript SG stands for.

**=> The same terminology and acronym has been used for the ground strike point density as defined in IEC 62858 Ed. 2. Indeed, one may wonder whether this abbreviation covers the name correctly.**

5. Figure 1, please label stroke no in (b), as you did for (a).

**=> This will be adjusted in an updated version of the manuscript.**